# Image-guided optogenetic spatiotemporal tissue patterning using $\mu$PatternScope

Sant Kumar [1,4], Hannes M. Beyer [2,4], Mingzhe Chen[1], Matias D. Zurbriggen [2,3] ✉ & Mustafa Khammash [1] ✉

In the field of tissue engineering, achieving precise spatiotemporal control over engineered cells is critical for sculpting functional 2D cell cultures into intricate morphological shapes. In this study, we engineer light-responsive mammalian cells and target them with dynamic light patterns to realize 2D cell culture patterning control. To achieve this, we developed $\mu$PatternScope ($\mu$PS), a modular framework for software-controlled projection of high-resolution light patterns onto microscope samples. $\mu$PS comprises hardware and software suite governing pattern projection and microscope maneuvers. Together with a 2D culture of the engineered cells, we utilize $\mu$PS for controlled spatiotemporal induction of apoptosis to generate desired 2D shapes. Furthermore, we introduce interactive closed-loop patterning, enabling a dynamic feedback mechanism between the measured cell culture patterns and the light illumination profiles to achieve the desired target patterning trends. Our work offers innovative tools for advanced tissue engineering applications through seamless fusion of optogenetics, optical engineering, and cybernetics.

During embryonic development, the spatial distribution of signals orchestrates the intricate processes involved in morphogenesis, guiding the formation of various shapes and structures[1]. Morphogenesis, in turn, modulates several pivotal biological processes, including cell growth, migration, proliferation, differentiation, and death across diverse tissues and organisms[2–4].

Recent years witnessed notable strides in tissue engineering and synthetic biology, intending to enhance our comprehension of the involved processes and capabilities to craft artificial tissues. These advances have birthed innovative methodologies and approaches to direct morphogenesis and pattern formation[5–10]. In natural settings, a combination of intra- and extracellular stimuli, genetic programs, and spontaneous stochastic events trigger the activity of signals initiating morphogenetic events. Yet, steering morphogenesis in vitro, such as in artificial tissue models, demands specialized toolkits that can induce these events with precision. Up to now, exogenous supplementation of

chemical stimuli offers some control over cell differentiation and morphogenesis but often lacks the finesse of spatial and temporal definition[11]. Thus, recent experimental endeavors began merging a plethora of innovative techniques to ensure precise modulation of developing tissues. Prominent among these technologies are mammalian cell genome engineering and chemically-inducible promoters. These advancements have catalyzed progress in the field, paving the way for engineered tissues that, upon stimulation, are programmed to execute specific biological functions. Such functions encompass synthetic morphogenesis actions like mammalian cell adhesion and cell death, achieved through the selective induction of transgene expression[12].

The advent of optogenetic techniques has revolutionized our ability to intervene in biological functions through optically-defined stimulation, giving birth to the field of optogenetics[13,14]. Among various external stimuli, light stands out due to its unparalleled precision,

[1]Department of Biosystems Science and Engineering (D-BSSE), ETH Zürich, Klingelbergstrasse 48, 4056 Basel, Switzerland. [2]Institute of Synthetic Biology, Heinrich-Heine-University Düsseldorf, Universitätsstrasse 1, D-40225 Düsseldorf, Germany. [3]CEPLAS - Cluster of Excellence on Plant Sciences, Düsseldorf, Universitätsstrasse 1, D-40225 Düsseldorf, Germany. [4]These authors contributed equally: Sant Kumar, Hannes M. Beyer. ✉e-mail: matias.zurbriggen@uni-duesseldorf.de; mustafa.khammash@bsse.ethz.ch

offering spatial, temporal, and quantitative control − making it ideal for in vitro tissue engineering and the study of tissue morphogenesis. This precision outpaces alternatives like chemical stimulation, especially when creating intricate patterns across engineered tissues. Advancements in optogenetics have showcased light as an effective tool for controlling a broad palette of cellular processes including gene expression[15], protein-protein interactions[16], and other applications[17]. Numerous optogenetic tools, especially those tailored for mammalian cell systems, are becoming game-changers in developmental biology and molecular biology research[18–20]. Optogenetics has enhanced our understanding of pivotal processes such as GTPase Rac-regulated epithelial cell movement in vivo[21] and developmental gene networks[22]. Initial efforts in directing morphogenetic events optogenetically in mammalian cells have yielded breakthroughs like controlling cell motility[23], manipulating adherent junctions[24], and modulating cell contractility or apical constriction[25,26].

For precise spatiotemporal optogenetic induction of morphogenetic effects, both an engineered optogenetic gene circuit in cells and a high-quality light delivery system, ranging from 2D to 3D regimes, are essential. Although tissues mainly develop in 3D, analyzing and directing synthetic morphogenetic patterns in 2D cell populations remains crucial for our comprehension of the processes and can be effective for both 2D and 3D tissues. Among techniques for light delivery in 2D, microscope-coupled projector-based platforms can stimulate individual cells while facilitating real-time microscopic evaluation, bridging computer software and cellular activities[27]. These systems often modify commercial projectors, necessitating substantial modifications, which can lead to optical distortions[28,29]. Few designs, such as Zhu et al.[30], have adopted a fully customized approach, integrating a dedicated digital micromirror device (DMD) used in digital light processing (DLP) projectors[31]. Still, these designs[30,32] are complex, hard to replicate, and often possess application-specific software. At the same time, dedicated commercial systems[33–35] are costly and have restrictive licenses, hindering adoption and customization in biology labs.

In this study, we introduce an optogenetic framework that combines engineered light-sensitive cells with innovative hardware and software for microscopic patterned light illumination. This integrated system enables precise spatiotemporal dynamic control over the morphology of engineered mammalian cells. At the core of this technology is the custom-built DMD-based system: $\mu$PatternScope ($\mu$PS). This apparatus seamlessly integrates with microscope systems, enabling simultaneous pattern illumination and real-time microscopic analysis. Assembling the modular $\mu$PS components is straightforward, requiring only basic tools and no specialized expertise. Notably, the entire hardware setup costs -USD 7-8k. Augmenting this hardware is an accompanying versatile $\mu$PS software suite. It offers comprehensive control over microscope functions and peripherals while managing the $\mu$PS hardware. With the dual capability of microscopic evaluation and optogenetic stimulation, the framework introduces a "cybergenetics" feedback control[27,36]. This feature allows real-time feedback control of cellular processes by adjusting light stimulation based on the observed in vivo outcomes, aiming for desired outputs such as set-point tracking. Additionally, the $\mu$PS software incorporates modules for single-cell segmentation and tracking, furnishing tools to measure, monitor, and regulate individual cell responses in optogenetic experiments. Its modular and open design ensures that $\mu$PS can easily accommodate further software enhancements.

In addition to a suitable illumination source, the optogenetic stimulation of mammalian tissues with high spatiotemporal resolution requires genomic cell engineering to imbue the cells with light-sensory properties. While transient transgene delivery can be adequate for developing optogenetic tools or affecting individual cells, achieving uniform responses across 2D or 3D cell tissues demands comprehensive genomic engineering and clonal selection[37]. For mechanisms like

cell division control and programmed cell death − crucial for cell homeostasis − it is imperative that every cell responds to stimulation; otherwise, non-responsive cells would disrupt the desired shapes, for instance, through unchecked growth. To address this, we developed an engineered mammalian cell line (ApOpto) harboring a genetic circuit for the rapid induction of apoptosis by blue light, the most relevant form of programmed cell death during the development of organisms.

To demonstrate the capabilities of our integrated system, we utilized the $\mu$PS framework to trigger the induction of apoptosis by pattern projection in a 2D population of ApOpto cells proliferating under a microscope. We then realized feedback control capability via the developed framework, and successfully implemented a 'tic-tac-toe' game played by two virtual computer players. Here, the players execute their moves by drawing shapes onto a 2D lawn of ApOpto cells by local induction of apoptosis through the projection of a corresponding pattern (here: cross or circle patterns). The experiment demonstrates the capability of our $\mu$PS framework to implement complex microscope control routines, and to realize feedback control experiments, together with optogenetically inducing morphogenetic patterning in mammalian cell cultures.

## Results
### $\mu$PatternScope framework design - hardware & software
The shortcoming of available solutions for implementing optogenetic projection-under-the-microscope in existing biology labs inspired us to develop the $\mu$PS framework. For the hardware design, we focused on an easy-to-assemble approach that does not necessitate expert knowledge (Fig. 1a, b; Supplementary Fig. S2a). Central to our design is a 0.65-inch diagonal DMD with over 2 million tilt-capable micromirrors (with 7.56 $\mu$m pitch) yielding high 1080p resolution mirror patterns. As previously stated, DMDs are electrically addressed spatial light modulators and key components of commercial wall projectors[31]. We used an off-the-shelf "telecentric" optical engine to homogenize and guide the incident light emitted from a high-power LED to fall onto the DMD at a desired angle. Based on the tilt state (ON or OFF state) of an individual micromirror within the DMD array, the optical engine would guide the mirror-reflected light either through the optical axis of the microscope episcopic-illumination path (ON-state) or away from it (OFF-state), as illustrated in Fig. 1a. Each micromirror oscillates between these two tilt states with fast pulse width modulation (PWM), enabling user-defined variation in the projected light intensity. The PWM duty cycles of all the oscillating micromirrors thus determine the light pattern being forwarded toward the microscope. This light pattern then passes through a series of three lenses (Fig. 1a) before entering the episcopic-illumination port of the microscope. The intermediary optics in the microscope further guide the pattern toward the objective lens, which ultimately focuses the projection pattern onto the sample plane (Fig. 1b inset images; Supplementary Fig. S2b). The optical configuration and a representative light ray diagram are illustrated in Supplementary Fig. S1.

We integrated a liquid light guide (LLG)-based assembly to attach and conveniently exchange the LED light source to the optical engine. Many off-the-shelf multi-color light engines with LLG output, e.g. Spectra light engines (Lumencor), can also be attached to the $\mu$PS hardware allowing multi-color illumination options. The complete optical path assembly and the DMD chip mount to the microscope via readily available and easy-to-install mounting brackets and cage rods, compatible with standard optical breadboards, resulting in a compact integrated design (Fig. 1b, Supplementary Fig. S2a, Supplementary Movie 1).

Similarly, the $\mu$PS software (Fig. 1c) follows a modular architecture, with code routines implemented as intuitive MATLAB functions and scripts. Even though MATLAB is a proprietary tool, it is ubiquitously present in the academic environment and academic

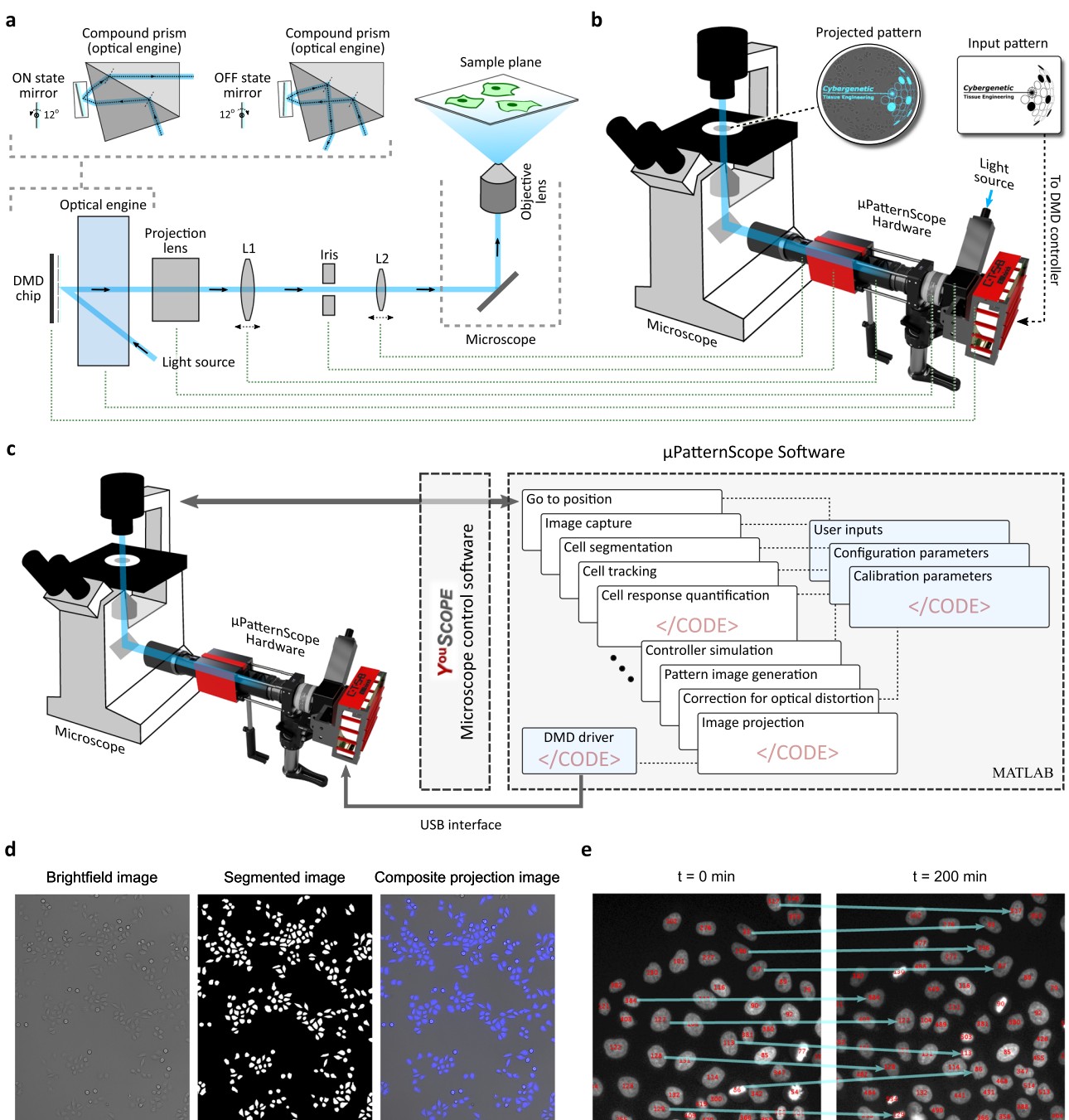

**Fig. 1 | μPatternScope (μPS) framework design - hardware & software.**
**a** Hardware design sketch. The μPS hardware design homogenizes light from an interchangeable LED light source and guides it (via an optical engine) toward a digital micromirror device (DMD). The reflected light pattern defined by oscillating ON and OFF tilt states of individual micromirrors passes a projection lens and then biconvex lenses (L1 and L2) before entering the episcopic-illumination path of the microscope. This set of lenses and intermediary optics focus the projected pattern onto the microscope sample plane via the objective lens (Supplementary Fig. S1). An iris aperture, mounted at an intermediate pattern image location between L1 and L2, assists in tuning the projection field of view under the microscope and aligning the optical path. **b** Computer aided design (CAD) model of the μPS hardware attached to a microscope. Inset figures show the projection of an input image pattern onto a population of HeLa cells placed under the microscope. The image is first sent to DMD controller, which sets the PWM duty cycle of oscillating DMD micromirrors to replicate the input pattern. This pattern is then focused onto the microscope sample plane as outlined in (**a**). **c** μPS software suite comprises several

MATLAB function and class modules for diverse tasks such as controlling the microscope, generating/correcting input projection image, controlling μPS hardware, etc. Integration of YouScope microscope control software[38] provides a GUI for live-imaging and microscopy observations. **d** Cell-segmentation operation using fastER tool-set[40]. Left, brightfield image (10X magnification) of HeLa cells proliferating under a microscope. Center, segmentation using fastER generates a mask pattern image. Right, blue light projection of the segmentation mask onto cells via the μPS hardware at single-cell resolution. Scale bar, 200 μm. **e** Cell-tracking algorithm, adapted from ref. 29, further expands the μPS framework to observe and control temporal dynamics of single-cell responses. Fluorescence images (20X magnification) of two time points from a time-lapse microscopy experiment of proliferating HeLa cells with miRFP670 tagged nuclear marker are shown. Individual nuclei were tracked over the duration of the experiment (indicated by identifier numbers), demonstrating cell-tracking capability of the μPS framework. Scale bar, 10 μm.

versions exist, making it a suitable rapid-prototyping design choice with no further required compilers or interpreters. We provide a set of useful functions to facilitate developing scripts for automated experiments. These functions cover a wide range of microscopy tasks, for example, capturing microscopy images, moving the microscope stage to a desired location, etc. Our software uses YouScope[38], an open-source microscope control software, to get access handles for different microscope peripherals and their functions. YouScope also provides a user-friendly graphical user interface (GUI) to perform fast microscopy live-imaging and initial image captures, required for adjusting experimental setups.

Furthermore, we developed a dedicated software module to send arbitrary pattern images via the DMD controller board onto the microscope sample plane. The module employs a previously-developed computer-to-DMD communication driver[39]. The $\mu$PS hardware ensures uniform (Supplementary Fig. S4) full field-of-view pattern projection under the microscope with limited optical or alignment distortions (Supplementary Fig. S3). In addition, we have developed a calibration code routine to compute the mapping between an input pattern image (DMD pixels) to the actual projected pattern imaged under the microscope (camera pixels). These mapping parameters adjust and correct a given input pattern image before sending it to the DMD controller board, resulting in the desired corrected pattern projection (Supplementary Fig. S5).

To fully utilize the spatial precision of the $\mu$PS hardware, we integrated a cell-segmentation tool into the software suite which allows, for example, illuminating individual cells within a culture while respecting the morphology of cells. The machine learning-based segmentation tool fastER (fast segmentation with extremal regions)[40] gathers candidate regions from a microscopic image via thresholding, extracts smartly-chosen 9 texture and shape features from those candidate regions, and then uses a trained support vector machine (SVM) to estimate the likelihood of the regions to represent a cell. Finally, the module defines individual cells as an optimal set of non-overlapping regions. The fastER code has previously been benchmarked as one of the most efficient cell segmentation tools[40]. It achieves state-of-the-art segmentation quality while still being orders of magnitude faster compared with other contemporary methods such as ilastik[41] and the neural network-based U-Net[42] tool. fastER also provides a user-friendly GUI for manual annotation and fast training. Figure 1d shows an example of a cell segmentation task (on a captured brightfield image of HeLa cells under the microscope) performed via $\mu$PS routine. The three images from left to right represent the acquired microscope image, the segmentation mask computed by the fastER tool within the $\mu$PS framework, and the patterned blue light image projected back onto the cells.

We further implemented an algorithm into the software suite for real-time tracking of cells. We incorporated a previously described method to observe the temporal dynamics of single-cell responses in time-lapse experiments[29]. A representative cell-tracking demonstration with a time-lapse imaging experiment is shown in Fig. 1e.

The high degree of modularity represents a key feature of the $\mu$PS framework. Both hardware and software modules may easily be extended or modified. The above-mentioned U-Net[42] cell-segmentation tool, for example, can be seamlessly integrated into the $\mu$PS framework. With the attached microscope system offering real-time observation of in vivo target processes, the $\mu$PS framework also implements in silico feedback control operation, which we demonstrate later in this paper. We envision that the $\mu$PS framework, in its current form, has the potential to provide a basis framework for further expansion toward a myriad of optogenetic patterning applications and studies.

## Optogenetic ApOpto gene switch for the induction of apoptosis with blue light

The induction of apoptosis, a form of programmed cell death, obeys a tight regulation of molecular actions. Cells may initiate apoptotic cell death either by integrating intrinsic, non-receptor-mediated signals such as DNA damage and other forms of stress or extrinsic signals mediated by transmembrane receptors like those belonging to the tumor necrosis factor (TNF) receptor gene superfamily[43]. Both pathways execute a proteolytic cellular cascade involving caspases, cysteine proteases engaged as the major coordinators of apoptosis. Among the various caspase isoforms identified in humans to date, initiator caspases (caspase-2,-8,-9,-10) and inflammatory caspases (caspase-1,-4,-5,-12) integrate early signals, eventually resulting in the irreversible induction of apoptosis through proteolytic processing of executioner caspases (caspase-3,-6,-7)[44]. Caspases are synthesized as zymogen precursor molecules called procaspases. Despite being present in a cell, procaspases remain catalytically inactive, but can readily be activated through processing by caspases upstream of the cascade. The specific proteolysis of a procaspase separates its subunits to allow the assembly of a catalytically active complex (Fig. 2a).

In order to induce apoptosis in mammalian cells upon optogenetic stimulation, we engineered a cell line (ApOpto) to synthesize a constitutively active form of the human caspase-3 (revCASP3, see below) upon blue light illumination. We selected caspase-3 as a molecular target, which acts as a downstream executioner caspase in the apoptotic proteolytic cascade with intrinsic signal amplification due to its enzymatic activity. This achieves a relatively fast response to light stimulation. For this aim, we resorted to an optogenetic gene switch that regulates specific transcriptional activities in cells upon blue light illumination (Fig. 2b). We have recently established vector sets for variants of blue and red light-sensitive optogenetic gene switches compatible with transposase-mediated genomic cell engineering, which synthesizes the genetic components from a few genomic copies in mammalian cells[37]. In fact, among the repertoire of available optogenetic tools (refer to optobase.org for an overview), optogenetic gene switches provide the highest degree of flexibility, because they separate the optogenetic component from the biological target. Rather than (re-)engineering the light-responsive and the effector components for every application, gene switches may require only minor adaptation, such as the exchange of the target gene of interest to be expressed under the corresponding illumination regime.

We used LOVpep as the optogenetic gene switch, a blue light-sensitive split transcription factor element based on the photo-switchable Light-Oxygen-Voltage domain (LOV2) from *Avena sativa* phototropin 1. LOVpep has previously been developed for mediating protein-protein interactions by controlling the solvent exposure of an engineered peptide by blue light to access engineered variants of the Erbin PDZ domain as binding partners (a technology also referred to as TULIPs, Tunable, light-controlled interacting protein tags)[45]. The LOVpep/ePDZb couple has further been derived as a gene switch for controlling transcription in mammalian and plant cells[37,46,47]. We used the erythromycin repressor protein E to tether LOVpep to an inducible DNA promoter element (Fig. 2b, c). Here, blue light exposure recruits a potent transcriptional activation domain, VP16, via ePDZb to induce the expression of *revCASP3* for the induction of apoptosis. The addition of the antibiotic erythromycin causes an allosteric release of E from the DNA to effectively shield cells from undergoing apoptosis, even despite being activated by blue light, and hence serves as phototoxicity control. We flanked the genomic components by transposase-specific terminal repeats on two individual vectors, together allowing the genomic transposition of the required sequences and the selection of mammalian cell populations harboring the optogenetic ApOpto gene switch controlling the expression of *mCherry* along *revCASP3* upon blue light illumination (Fig. 2c). For engineering a constitutive active caspase-3, we followed a previously proposed design strategy which reverts the order of subunits of the procaspase with the small (p12) subunit preceding the large (p17) subunit (revCASP3, Fig. 2c)[48]. The propeptide serves as a flexible linker, facilitating folding into an active conformation upon synthesis. In a wild-

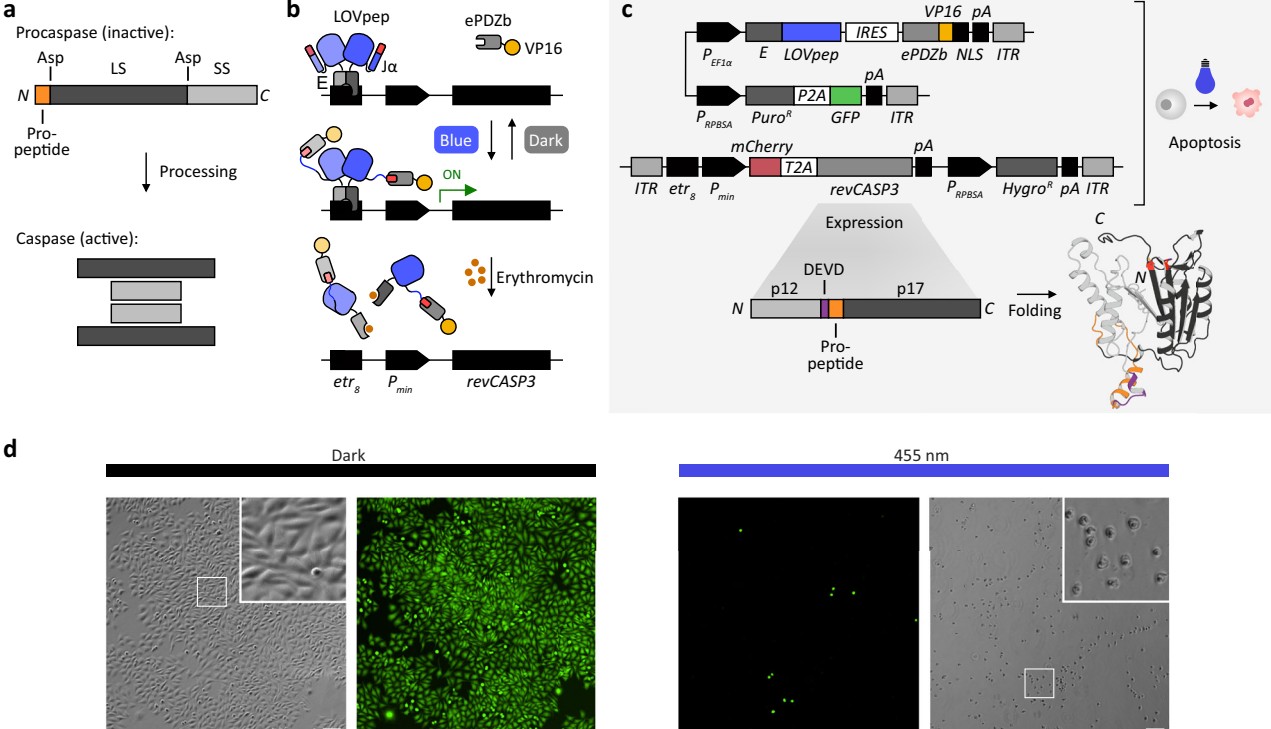

**Fig. 2 | Design and genomic engineering of ApOpto cells for optogenetic induction of apoptosis. a** Caspases mature from inactive precursors (procaspases) through proteolytic processing at Asp residues to remove a leading propeptide and separate the protein into a large (LS) and small (SS) subunit, forming a catalytically active caspase complex. **b** The LOVpep/ePDZb photoswitch utilizes the erythromycin repressor protein E for tethering LOVpep to an inducible promoter. Blue light induces a conformational change in LOVpep to expose an engineered peptide (red) to initiate a protein-protein interaction with ePDZb, leading to the recruitment of the transcriptional activation domain VP16 and induction of *revCASP3* transcription. **c** Genetic vector design for the generation of genomically engineered ApOpto cells using transposases and puromycin and hygromycin selection. The

target gene for optogenetic induction, *revCASP3*, has a reversed order of subunits, with the small p12 preceding the large p17 subunit and the propeptide. Upon synthesis, revCASP3 immediately folds into an active conformation. The presence of the caspase-3-specific DEVD sequence supports autocatalytic processing. revCASP3 model generated using RosettaFold and PyMOL[71,72]. For abbreviations, see Methods. **d** CHO-K1[ApOpto] cells (Clone # 4, Supplementary Fig. S7) incubated for 24 h either in the dark (left), or under 455 nm blue light (right) with an intensity of 10 $\mu$mol m$^{-2}$ s$^{-1}$. GFP and brightfield images with magnification are shown. See Supplementary Fig. S7 for the same image together with all isolated clones. Scale bar, 100 $\mu$m.

type procaspase configuration, the domain arrangement effectively restricts their functional assembly. The sequence connecting the subunits in revCASP3 further harbors a DEVD (amino acid sequence) caspase-3-specific target sequence to promote autocatalytic processing.

We generated stable transformants carrying the ApOpto gene switch using different mammalian cell lines (HEK-293, HEK-293T, CHO-K1, Supplementary Fig. S6a–c). In each cell line, individual cells clearly showed unique responses to 24 h periods of blue light illumination, presumably due to the stochastic transgene integration events. Blue light exposure induced *revCASP3* expression and apoptotic cell death in individual cells in CHO-K1 and HEK-293T-derived cultures, despite a mutation within the *mCherry* gene, which we identified after initial successful tests (see Methods). HEK-293 cultures appeared much more sensitive to the induction. Here, the majority of cells detached from the substrate in illuminated samples. However, incubation of the cultures in the dark, or blue light with erythromycin supplementation, effectively protected the cells from undergoing apoptosis, suggesting that the observed cell death indeed resulted from the optogenetic induction of apoptosis and was not caused by phototoxicity. We then used the HEK-293 culture to generate 3D spheroid tissues which likewise suffered cell death under blue light but remained viable in the dark or when protected by erythromycin, suggesting that blue light sufficiently penetrates 3D tissues to induce the gene switch (Supplementary Fig. S6d). However, for microscopic studies at a higher magnification and the use of spatial optogenetic stimulation with complex

patterns in 2D, we considered the CHO-K1 culture as best suited due to its homogeneous growth. To obtain a uniformly responding culture, we performed a clonal selection by random isolation of 29 individual single-cell clones (Supplementary Fig. S7). Among those, we found a variation in cell death induction strength upon exposure to blue light. We chose Clone #4 for further experiments, which we refer to as CHO-K1[ApOpto] (Fig. 2d). When grown in the dark, CHO-K1[ApOpto] cells displayed a viable phenotype reminiscent of the parental cell line and a constitutive GFP signal, indicating the genomic integration of the LOVpep/ePDZb photoswitch. Under blue light illumination, however, all cells either detached from the substrate, or showed characteristic morphological changes including fragmentation into apoptotic bodies, indicating progressive apoptosis along with an overall loss of GFP fluorescence.

**Spatial apoptosis pattern induction**

We next tested the spatial induction of apoptosis by projecting a blue light pattern onto a 2D culture of CHO-K1[ApOpto] cells growing under the microscope. Using the $\mu$PS framework, we generated a finger-like illumination pattern spanning the microscopic view field through a 4X objective lens with 1.5X additional manual magnification (Fig. 3a). Here, we defined "fingers" as dark regions while exposing the surrounding area to blue light. Within 16 h of continuous projection via $\mu$PS, we indeed observed the induction of cell apoptosis within the illuminated regions, evident in both brightfield and relevant fluorescence images in Fig. 3a, b. Here, SYTOX Blue, which we used as a stain

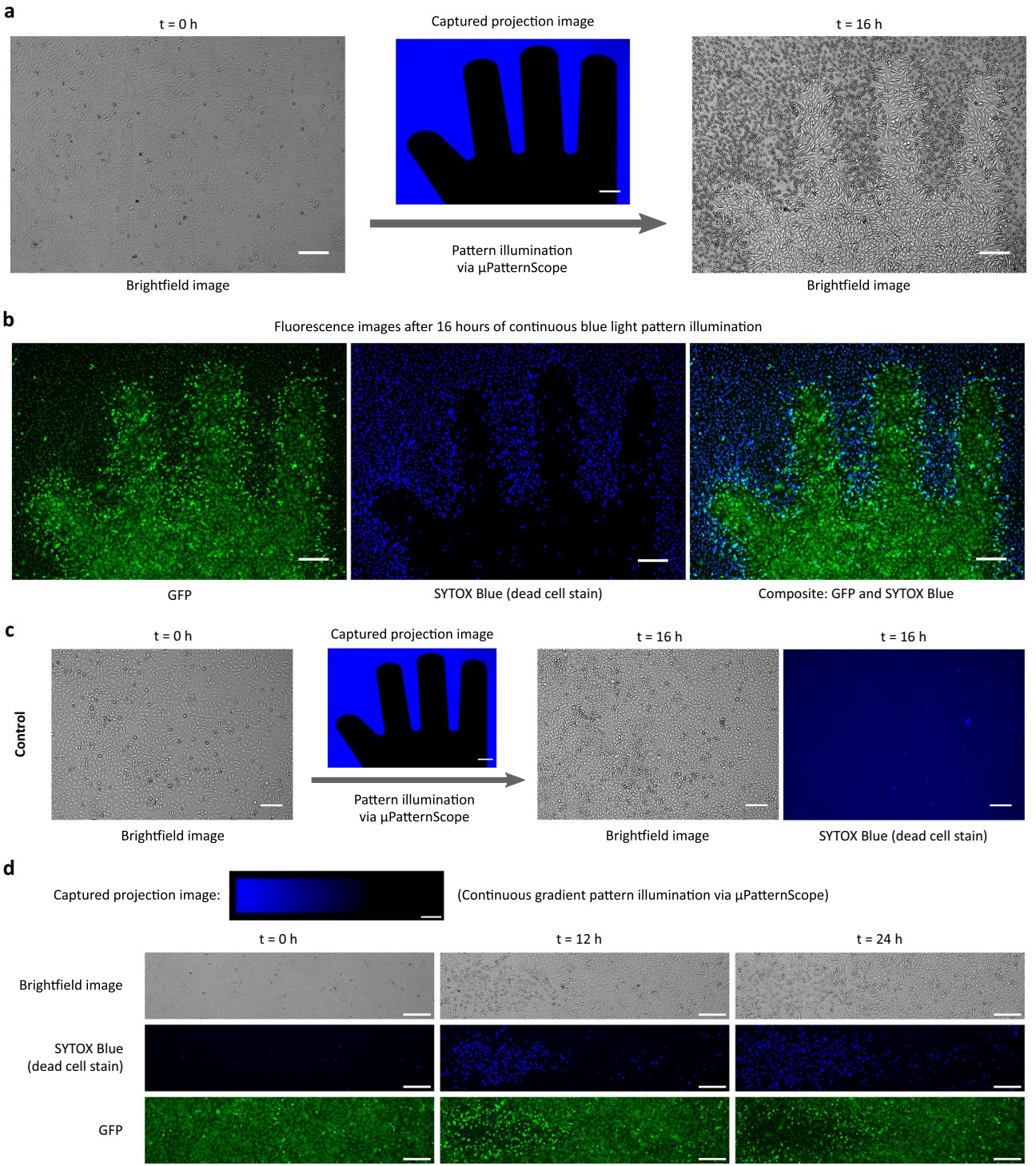

**Fig. 3 | Optogenetic induction of apoptosis in a spatial pattern. a, b** Spatial induction of apoptosis on a cell sheet of engineered CHO-K1^(ApOpto) cells by continuous illumination with a blue light pattern for 16 h via the µPS framework. **c** Control experiment as in (**a, b**) but using the parental CHO-K1 cell line lacking the engineered optogenetic apoptosis circuit (Fig. 2). No spatial cell death could be observed, indicating that the used light intensity is not phototoxic. **d** Continuous blue light intensity gradient illumination over CHO-K1^(ApOpto) cells via the µPS. Scale bar, 200 $\mu$m. Blue light illumination irradiance (**a**–**c**, max. irradiance in **d**), ~ 175 $\mu$W/cm².

for the nuclei of dead cells, indicated a far progression of apoptosis (Fig. 3b (center)), while cells in the dark regions remained viable and possessed a regular GFP signal (Fig. 3b (left)). Cells at border regions, separating illuminated and dark areas, appeared in both the SYTOX Blue and the GFP channel (Fig. 3b (right)). These short-range edge effects presumably result from cells migrating from the dark into the illuminated region, causing a delayed induction of apoptosis

accompanied by an enhanced GFP signal due to the onset of apoptosis associated with morphological changes such as cell shrinkage. Continuous cell proliferation exclusively in dark regions of confluent living cells promotes unidirectional movement. In a control experiment using the parental CHO-K1 cell line, the projection of the blue light pattern did not cause a visible degree of cell death (SYTOX Blue positive cells) within the illuminated area, as only very few signals

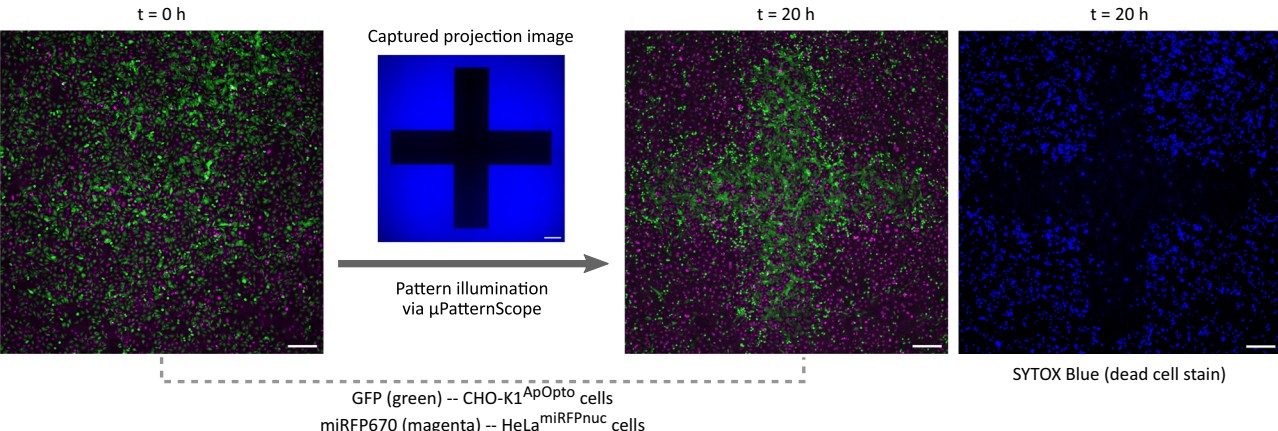

t = 0 h

Captured projection image

t = 20 h

t = 20 h

Pattern illumination
via μPatternScope

SYTOX Blue (dead cell stain)

GFP (green) -- CHO-K1^ApOpto cells
miRFP670 (magenta) -- HeLa^miRFPnuc cells

**Fig. 4 | Spatial apoptosis patterning in a co-culture.** Spatial induction of apoptosis in a co-culture of engineered CHO-K1^ApOpto (green) and HeLa^miRFPnuc (magenta) cells proliferating under the microscope by patterned illumination via μPS. In blue light-exposed areas, CHO-K1^ApOpto cells induced apoptosis and acquired a SYTOX Blue positive signal, while HeLa^miRFPnuc cells lacking the optogenetic apoptosis circuit proliferated throughout the view field. Fluorescence images at the start and after 20 h of patterned projection are shown. Scale bars, 200 μm. Blue light illumination irradiance, ~195 μW/cm².

emerged with random spatial distribution throughout the image (Fig. 3c). This experiment confirmed that the used light intensity is not phototoxic.

Next, we tried to qualitatively understand the temporal dynamics of the optogenetic apoptosis induction. We illuminated CHO-K1^ApOpto cells with a continuous blue light intensity gradient and captured time-lapse microscopy images every 30 min (Fig. 3d and Supplementary Movie 2–4). After 24 h of illumination, the degree of apoptosis induction correlated well with the intensity profile of the gradient, with the strongest SYTOX Blue signal observed in areas of the highest blue light intensity. In this region, the first SYTOX Blue positive cells appeared as early as 4 h. Regions of lower intensity induced the onset of apoptosis significantly slower, and low-intensity regions only showed a negligible degree of cell death, suggesting that CHO-K1^ApOpto cells tolerate exposure to blue light up to a certain threshold (as observed in Supplementary Fig. S8). The activation rate of LOVpep depends on the light intensity and suffers rapid dark reversion ($t_{1/2} = 17\,s$)[46]. Our experiments show that even though constant illumination causes productive gene transcription, one can also employ a pulsatile illumination strategy with suitably higher blue light illumination intensities to achieve similar results as shown in Supplementary Fig. S9.

In mixed cultures consisting of optogenetically sensitive and insensitive cell populations, activation of apoptosis using spatially-defined pattern projection can selectively remove cell types (Fig. 4). In a co-culture consisting of CHO-K1^ApOpto (green) and HeLa^miRFPnuc cells expressing a nuclear miRFP670 marker (magenta), we successfully induced apoptosis in a defined area exclusively in CHO-K1^ApOpto (blue) while not affecting the second cell population.

The series of experiments suggests that a combination of engineered hardware and software, together with a strategy for optogenetic cell engineering as compiled in the μPS framework, enables high-precision optogenetic pattern modulation of synthetic mammalian tissues.

## A light-controlled 'tic-tac-toe' game demonstrates closed-loop feedback decision-making

Computer-assisted programming of biological systems is challenging due to the limited possibilities of interfacing cellular systems with computers[28,49,50]. Genetically engineering cells with optogenetic capabilities generates one such interface, as light stimuli effectively transmit wireless signals to target cells, enabling their dynamic regulation via software algorithms. The last couple of decades witnessed rapid advancement in computing technologies relevant to biological

sciences, yielding a repertoire of sophisticated analytical methodologies to study molecular mechanisms in cellular systems. These developments increasingly involve hybrid technologies that make it possible to interface biological and computer systems— a developing area that has become known as "cybergenetics"[36,51–60]. In this work, cellular processes interface with a controller algorithm that integrates real-time monitoring of biological samples with in silico simulation and the automated control of executive cell-compatible stimuli. Suitable sensing and actuation interfaces enable the controller to monitor and quantify the target in vivo process and adjust stimuli in real-time. Our μPS framework provides a suitable platform to conduct optogenetic feedback control studies by quantitatively monitoring the cellular target behavior from microscopy images, executing controller-based computer simulations using information acquired in real-time, and then adjusting the target behavior as desired via the optogenetic regulation. Together with a suitably engineered cellular system such as the CHO-K1^ApOpto cell line, the framework would hence enable closed-loop feedback control operations in mammalian cells, which has not yet been achieved to date.

To demonstrate the application of μPS framework in realizing software-based control over a biological design, we implemented a dynamic and automated tic-tac-toe game with a 2D culture of CHO-K1^ApOpto cells serving as the playfield (Fig. 5). Two virtual computer players participate by alternately drawing cross and circle patterns onto a 3 × 3 raster area, realized by the projection of the respective blue light patterns via the μPS hardware. The projected patterns manifest on the playfield grid by local induction of cell apoptosis. A player who executes three adjacent patterns in a straight line wins the match.

Our implementation builds upon an interplay of actions taken by three independent representative agents: the game handler governing the administration of rules, computer player A (executing circle patterns), and computer player B (executing cross patterns). The match is initialized by assigning nine non-overlapping positions onto a 2D cell culture under the microscope, designated as P1 to P9 (Fig. 5a). The game handler maintains a virtual tic-tac-toe grid whose 9 playable squares are mapped to the 9 pre-defined locations (P1 to P9) over cells. Computer players A and B take turns to execute circles and crosses. Once a player makes a circle or cross move on the virtual grid, the game handler then projects a circle or cross light pattern (Fig. 5a) at the corresponding mapped location under the microscope for 5 h, which is just enough time for the CHO-K1^ApOpto cells to undergo observable patterned apoptosis.

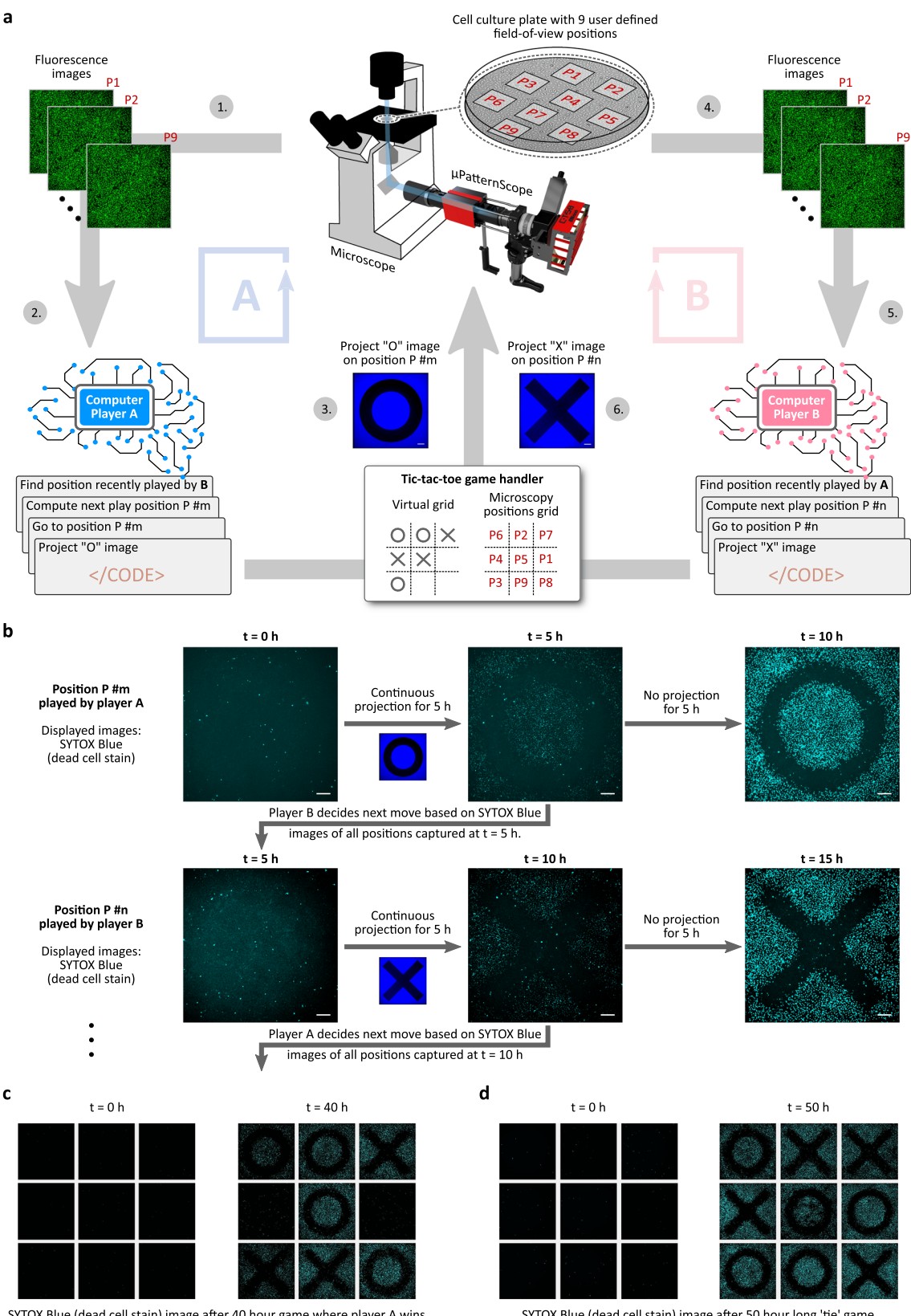

In a logical flow of the game (Fig. 5a, b), after every turn (5 h continuous pattern projection), the game handler captures fluorescence images (measurement) of all locations (P1 to P9). Through these images, the current-turn player identifies the previous move made by the other player by analyzing the SYTOX Blue pattern arising from the spatial optogenetic induction of apoptosis through thresholding and image processing techniques. The player then performs smart computation (adapted from ref. 61) in order to execute the next move on the updated tic-tac-toe grid. Based on this move, the game handler then sends the command to illuminate the circle or cross pattern onto the correspondingly mapped position for 5 continuous hours. After this, the game handler and the other player

**Fig. 5 | Implementation of an "apoptotic" tic-tac-toe game using the *μPS* framework and CHO-K1^ApOpto cells.** Two virtual computer players (CPs) participate in a conventional tic-tac-toe game using a sheet of proliferating CHO-K1^ApOpto cells as the game arena. Players execute moves by projecting circle and cross patterns of light onto the cells to induce the visual onset of apoptosis at 9 dedicated locations (P1 to P9) on the arena. **a** 9 non-overlapping microscopy locations, each serving as a unique playable position, are selected in a well-plate containing CHO-K1^ApOpto cells placed under the microscope. A representative game-flow proceeds as: 1. Acquisition of fluorescence (SYTOX Blue stain) images of the 9 pre-defined microscopy positions, 2. CP-A (here playing circle patterns) determines the position recently played by CP-B based on the analysis and processing of the images and computes the next move, that is, the position P #m to play with a circle pattern. 3. Position P #m is then illuminated with a circle pattern. 4. Acquisition of fluorescence images at

all 9 positions. 5. CP-B (here playing cross patterns) now determines the move previously played by CP-A and decides on position P #n to play with a cross pattern. 6. Position P #n is then illuminated with a circle pattern. A full match results from iterative repetitions of steps 1 to 6 until one of the players wins or the game ends in a tie. **b** At every turn, the chosen positions by the players are continuously illuminated with a circle (or cross) pattern for 5 h. Fluorescence images of all 9 positions are captured and serve the other player CP-B (CP-A) to decide on the next move. Therefore, each turn takes 5 h. For displaying the final results (see **c, d**), another image is captured after an additional 5 h without illumination has passed. Scale bars, 200 *μm*. **c** Results of a 40 h game experiment, where CP-A (playing circle patterns) won. **d** Results of a 50 h "tie" game experiment. Blue light illumination irradiance, ~ 260 *μW/cm²*.

follow the same measurement-computation-illumination steps for the next turn.

At each turn, to terminate the match, the game handler evaluates whether any of the players has already won the match (three circles or crosses in a straight line on the grid, Fig. 5c)). Otherwise, the game proceeds until all 9 positions have been played (Fig. 5d).

With the given execution intervals on a 3 × 3 grid (5 h pattern projection at every turn with an additional acquisition of the final pattern after 10 h), a complete match may take up to 50 h (45 h match with an additional 5 h for capturing the last image, Fig. 5b). To compensate for the overgrowth of cells throughout the match, we implemented a dynamic mapping routine and seeded cells at different densities. This allowed us to assign the initially played locations in the tic-tac-toe grid to higher confluent locations, and then dynamically map to lower levels of confluence over the progression of the game.

The implementation of the tic-tac-toe game requires the fine interplay of a series of technologies: (i) an automated spatiotemporal execution of cell morphogenesis (here: cell death through optogenetic induction of apoptosis), (ii) live monitoring of induction events, (iii) automated real-time image analysis, and (iv) feedback-based decision-making algorithms. A player comes to know about the move of the other player only by processing the fluorescence images captured at all nine mapped locations over cells, and then decides and executes the move (pattern projection) based on the processing of analysis data. Our results in Fig. 5c, d resulted in two possible outcomes, a 50 h long match which ended in a tie between the two computer players (Supplementary Movie 5, 6), and a shorter match where one player won after 35–40 h (Supplementary Movie 7, 8).

With this successful tic-tac-toe game implementation, we highlight some of the capabilities of the *μPS* framework, aimed towards the realization of image-based optogenetic feedback operations. The game implementation required interfacing the *μPS* with additional software modules specific to the game rules, demonstrating the modular and accessible architecture of the framework.

## Discussion

Optogenetic approaches to the study of complex biological systems both in vivo and in vitro bear a potential that in many aspects surpasses any alternative method, e.g. resorting to chemical stimulation. Cybergenetic regulation of biological tissues through computer-aided closed-loop control in particular benefits from the spatial and temporal precision intrinsic to optical stimulation technologies. The fine modulation of stimulation intensities, and the spatial definition of activation, combined with the straightforward possibilities to adjust light signals with software algorithms, make optogenetic technologies a perfect match for closed-loop circuitry coupled with automated microscopic analysis in real-time. To date, the repertoire of optogenetic switches with potential use in animal cell tissues has grown to a size worth indexing in databases[17] (www.optobase.org). However, utilizing their potential beyond proof-of-principle applications requires sophisticated accompanying technologies involving biological and

technical engineering. Such innovation may include genomic cell editing, but also illumination devices capable of generating the required precise optical stimuli. Unfortunately, the availability of these technologies currently represents a major bottleneck that significantly impedes the progress of the optogenetics field and discourages its adaptation in research areas that would in fact benefit from it.

In this work, we developed the *μPatternScope* (*μPS*) framework combining suitably-engineered optogenetic cells with an accessible and open-source hardware-software platform centered around DMD projection technology. The framework conveniently couples to a microscope to facilitate patterned light illumination of samples with simultaneous monitoring. The accompanying software automates complex sample illumination regimes and controls the operation of the microscope programmatically. Although the concept of microscope-coupled projection systems has previously been introduced[27], those systems either lack a dedicated design (microscope compatibility) causing undesired projection distortions or a flexible software framework for automating complex operations such as feedback control. Costly commercial solutions exist, however, their closed-sourced nature restricts modifications and generic adaptability in academic research environments. Our *μPS* framework fills these obvious gaps by combining projection hardware with a generic software architecture, featuring expandable modules for a wide range of operational tasks. The parts required to upgrade an existing microscope with the *μPS* framework range about ~ USD 7-8k, significantly lower (3–10 times) than the cost of commercial counterparts. Due to the use of off-the-shelf components, the hardware assembly and installation takes less than a day, and no expert technical skills (for detailed assembly instructions, please refer to the GitHub repository mentioned in the Methods section). Leveraging the scripting feature of YouScope, an open-source microscope control software[38], the *μPS* software module can control the microscope peripherals and their functions, for example, capturing microscopy images, moving microscope sample stage, etc.

We would like to highlight that the *μPS* framework facilitates sensing, actuation (illumination), and in silico computational operation at resolutions exceeding the dimensions of single cells, making it a suitable platform for studying diverse specimens beyond the examples demonstrated in this work. It maintains high-resolution illumination across different magnifications under the microscope (Supplementary Fig. S3), thus potentially allowing single-cell targeting for even smaller-sized cell types such as yeast. Furthermore, we integrated single-cell segmentation and tracking tools within the *μPS* software suite, providing the dynamic optogenetic control of individually-tracked cells as they migrate throughout the experiment, for example in cyberloop studies for rapid-prototyping cybergenetic single-cell controllers[58]. Additionally, the modular architecture of the *μPS* software allows integrating other existing software tools such as: DeLTA (automated cell segmentation, tracking, and lineage reconstruction)[62], Cheetah (a computational toolkit for cybergenetic control with a U-Net-based image segmentation system)[63], and MicroMator (a software for

reactive microscopy experiments)[64]. Depending on the targeting pattern and application requirements (for example, targeting single-cells in a confluent cell culture), one might need to tune light intensities within the projection pattern to ensure minimal illumination bleed-through between light and dark regions or between neighboring cells (Supplementary Fig. S3). Solutions, such as erosion of light intensities within the stimulation region[64], can be easily implemented in our software framework. The μPS developed in this work offers an accessible and cost-effective solution for dynamic, pattern projection-based optogenetic experiments. Its affordability coupled with highly adaptable open-source software, caters to a broad spectrum of specific requirements, emphasizing both illumination precision and automation.

We envision that our work will set an example methodology for realizing computer-based optogenetic stimulation of mammalian cell tissues. To demonstrate such an operation, we developed an optogenetic gene switch that induces the expression of a constitutively active variant of caspase-3, which acts as an executioner caspase with fast and signal-amplifying characteristics of apoptosis induction. We then genomically engineered the switch into different mammalian cells for spatially controlling the onset of apoptosis using blue light. The generic nature of the utilized switch potentially allows adaptation of the target effector gene to modulate a wide range of different morphogenetic or disease-relevant phenomena, relevant e.g., to the study of developmental biology or tissue engineering. Using the μPS framework, we optogenetically induced apoptosis in 2D tissue cultures with high spatial definition. Here, μPS served to characterize the resulting CHO-K1$^{ApOpto}$ cell line by simultaneously adjusting light intensity levels and exposure time measurements across a microscopic view field, suggesting the μPS framework as a suitable instrument for qualitative and quantitative studies on optogenetic tissues.

The future of optogenetic technologies will likely increasingly involve computer-aided systems to dynamically adjust optical stimuli while spatiotemporally regulating and observing the development of tissues. The μPS framework offers such a system design. It provides modules for both automatizing light pattern projection and image acquisition. To demonstrate the technology, we implemented a tic-tac-toe game played between two virtual players on a 2D culture of engineered CHO-K1$^{ApOpto}$ cells, forming the tic-tac-toe grid arena. Here, the μPS framework handled the game rules and interpreted the moves of the players by detecting circle or cross patterns drawn by spatial blue light induction of apoptosis onto a 3 × 3 grid of playable positions. This unique implementation involved image-based decision-making, as each player must process the microscopy-captured images to identify the move of the opponent player and decide on the position for the next move. Here, the open scripting nature of the μPS framework enabled such expansion, highlighting the very wide use scenarios of the here-developed technology. This cybergenetic framework, encompassing the live modulation of target cellular process in vivo influenced by in silico computations via suitable sensing and actuation interfaces, has been proclaimed as a vast utility in several emerging applications within the synthetic biology community. Potential applications range from bioproduction optimization[65] to various synthetic biology-inspired studies, including multicellular morphogenesis[60].

Our work examplifies precise spatiotemporal control of optogenetic cells in a dynamic environment using μPS framework and engineered CHO-K1$^{ApOpto}$ cells. Similar optogenetics-based methodologies could be applied to more advanced tissue engineering efforts by employing μPS to precisely orchestrate cell behavior, cellular signaling progressions, growth patterns, and interactions in real-time. Such capabilities enable the creation of complex tissue structures with specific cellular arrangements, facilitating research in tissue regeneration, disease modeling, and drug testing, where accurate tissue and cellular configurations are crucial. For example, advanced tissue engineering applications using the proposed technology might involve

creating vascular networks within engineered tissues, something that is currently difficult to achieve using existing technologies. The precise control offered by the approach could be used to pattern endothelial cells in a way that mimics the intricate network of blood vessels in tissues. This would be crucial for developing thicker, more complex tissue structures that require a blood supply for nutrients and oxygen, with great potential for advancing the field of organ regeneration and repair.

In summary, we introduced in this manuscript an innovative approach for the precision engineering of cell cultures. It combines optogenetic gene switches and relevant optical devices. The core technology is the μPatternScope (μPS), a modular framework for software-controlled projection of high-resolution light patterns onto microscope samples. This setup allows for dynamic, automated control of cellular processes, such as apoptosis, to create intricate 2D cell culture patterns. The manuscript showcases the capabilities of this cybergenetic patterning through various experiments, including an automated 'tic-tac-toe' game using a 2D cell culture. This work proposes innovative tools for advanced tissue engineering, integrating optogenetics, optical engineering, and cybernetics.

## Methods
### Molecular cloning
A gene encoding a constitutively active form of the human caspase-3 was generated by reversing the order of the small (p12) and large (p17) subunits including the propeptide, analogously to a previous report.[48] Additionally, a sequence encoding the DEVD caspase-3 substrate sequence was included preceding the propeptide to facilitate proteolytic self-maturation. The sequence encoding the red fluorescent protein mCherry together with the *Thosea asigna* virus 2A-derived self-processing T2A peptide was first PCR-amplified from plasmid pTREx-mCherry-2A-Cdh3[66] using the oligonucleotides oDD622 (5′-CTCCGC GGCCCCGGTACCGAATTCGAGCTCGCCCGGGCGCCACCATGGTGAG CAAGGG) and oDD623 (5′-CATCATCAACACCACTTGGGCCAGGATTCT CCTCC). The caspase-3 subunit p12 was then amplified from a vector derived from plasmid TU#817 (addgene #16084) and the p17 subunit together with the propeptide from a derivative of TU#818 (addgene #16085) using oDD626 (5′-CACGACGAGGTGGACGGCGGCAGCCC CATGGAGAACACTGAAAACTCAGTG) and oDD627 (5′-ATCATGTCTG GATCGAAGCTTGGGCTGCAGGTCGACTTAGTCTGTCTCAATGCCACA GTC), thereby including the DEVD-encoding sequence in the primer extension.

The DNA fragments were fused by PCR using oDD622 and oDD627, and then Gibson-cloned into *Xma*I and *Sal*I-digested pDD203[37] under the control of an inducible promoter regulated by synthetic transcription factors based on the erythromycin repressor protein E DNA-binding protein and vector elements for genomic transposition and hygromycin selection. However, we could exclusively isolate clones carrying mutations in the coding sequence, suggesting a toxic effect of the gene when transformed into *E. coli*, giving rise to the selection of mutated variants. Therefore, an SV40-derived intron was inserted into the p12-encoding sequence. The 5′ and 3′ regions of two clones carrying mutations at different positions were PCR amplified using oDD622 and oDD635 (5′-CATGTCATCATCAA-CACCACTTGG), and oDD634 (5′-GCGTGTCATAAAATACCAGTGGAG) and oDD627, respectively. The SV40 intron was PCR-amplified from plasmid pCMV(CAT)T7-SB100 (addgene #34879) using oDD636 (5′-TGGCCCAAGTGGTGTTGATGATGACATGGTAAGTTTAGTCTTTTTGT CTTTTATTTCAGG) and oDD637 (5′-GTCGGCCTCCACTGGTATTT-TATGACACGCCTAGAAGTAAAGGCAACATCCACTG). The fragments were fused by PCR using oDD622 and oDD627 and again cloned into *Xma*I and *Sal*I-digested pDD203, resulting in the final vector pDD265. For the generation of cell lines, the vector was used together with the previously-described plasmids pDD107 encoding the LOVpep/ePDZ

gene switch[37] and pCMV(CAT)T7-SB100 (addgene #34879) encoding the SB100X transposase (see below).

After the first successful tests of plasmid pDD265 encoding mCherry together with revCASP3, we surprisingly identified a single basepair deletion, two basepairs downstream of the start codon of the mCherry gene, eventually causing unproductive translation. However, an alternative start codon exists within the mCherry gene, located 22 basepairs downstream of the mutation. The sequence surrounding this ATG codon matches the Kozak consensus, and it has recently been shown that the codon leads to the translation of an alternative mCherry isoform both in prokaryotic and eukaryotic expression hosts. This isoform lacks nine N-terminal residues and is sometimes referred to as isoform V1[67]. As we identified erythromycin-sensitive blue light-induced expression of *mCherry* and induction of apoptosis, we confirmed the presence of this isoform and continued using it for the experiments in this study. The strength of induction in response to blue light illumination is likely compromised due to this effect, however, revCASP3 seems highly potent and the expression strength compensates during clonal selection which also eliminates cells with significant basal *revCASP3* expression by inducing apoptosis.

### Cell culture and cell line engineering (CHO-K1^ApOpto^)

Chinese Hamster Ovary cells (CHO-K1, DSMZ, Braunschweig, Germany, ACC 110) were cultivated in a humidified atmosphere at 37 °C, 5% $CO_2$ in Ham's F12 Medium (PAN Biotech, Aidenbach, Germany, cat. no. P04-14500) supplemented with 125 U mL$^{-1}$ penicillin, 125 U mL$^{-1}$ streptomycin (PAN Biotech, cat. no. P06-07100), and 10% fetal bovine serum (PAN Biotech, cat. no. P30-3602). Human embryonic kidney HEK-293 (DSMZ, ACC 305), HEK-293T (DSMZ, Braunschweig, Germany, ACC 635) were cultivated in Dulbecco's Modified Eagle Medium (DMEM, PAN Biotech, cat. no. P04-03550) with identical supplementation. For generating stable 293^ApOpto^, 293T^ApOpto^, and CHO-K1^ApOpto^ cultures using the SB100X transposase,[68] 350,000 HEK-293 or HEK-293T cells, or 250,000 CHO-K1 cells were seeded into two wells of a 6-well plate. The next day, the cells were transfected with plasmids pDD107 and pCMV(CAT)T7-SB100 (see above) at a ratio of 10:1 (w:w) using polyethylenimine (PEI, linear, MW: 25 kDa, Polysciences Inc. Europe, Hirschberg, Germany, cat. no. 23966 (1)) as described elsewhere[37]. The following day, puromycin (Thermo Fisher, Waltham, MA, US, cat. no. A1113803) was added to the cultures at a concentration of 4 μg mL$^{-1}$ (HEK-293), or 10 μg mL$^{-1}$ (CHO-K1, HEK-293T), the wells were combined, and then passaged for about two weeks until a stable GFP positive culture emerged without evident cell death caused by the antibiotic selection. The process was repeated with the obtained cells, this time using plasmid pDD265 instead of pDD107. 4 h post-transfection, the medium was exchanged with a fresh medium supplemented with 2 μg mL$^{-1}$ erythromycin (Sigma Aldrich, St. Louis, MO, US, cat no. E5389-1G) and 20 μM Pan-Caspase inhibitor Z-FAD-FMK (InvivoGen, San Diego, CS, US, cat. no. tlrl-vad). The next day, the medium was additionally supplemented with 400 μg mL$^{-1}$ (HEK-293, HEK-293T) or 500 μg mL$^{-1}$ (CHO-K1) Hygromycin B Gold (InvivoGen, cat. no. ant-hg-1). The treatment with Z-FAD-FMK and erythromycin was continued for one and two weeks, respectively. Hygromycin selection continued for two additional weeks before the cells were cryopreserved in liquid nitrogen for further use.

Single clones of CHO-K1^ApOpto^ cells were randomly selected by dilution and colony formation and cultivated in 24-well plates until all clones emerged in a stable culture in the presence of 2 μg mL$^{-1}$ erythromycin. After an additional passage in the absence of erythromycin, the clones were seeded in 24-well plates and subjected to optogenetic tests (see below). Some functional clones were further passaged and cryopreserved.

For optogenetic experiments using the μPS, CHO-K1 cells were seeded at appropriate densities (30,000–120,000 cells) into wells of a

24-well VisiPlate-24 Black (Perkin Elmer, Waltham, MA, US, cat. no. 1450-606) the day prior to the experiment.

### Cell culture and cell line engineering (HeLa^miRFPnuc^)

HeLa (ATCC, strain number CCL-2) cells were cultivated in a humidified atmosphere at 37°C with 5% $CO_2$ in Dulbecco's modified Eagle's medium (DMEM), high glucose, GlutaMAX Supplement, pyruvate (Gibco/Thermo Fisher Scientific, Waltham, MA, US, cat. no. 31966021), 10% FBS (Sigma-Aldrich cat. no. F7524-500ML, St. Louis, MO, US), 100 U/mL Penicillin-Streptomycin (Gibco/Thermo Fisher Scientific, Waltham, MA, US, cat. no. 15140122). Cell culture passage into a fresh Jet Biofil T25 flask (Witec, Sursee, CH, cat. no. TCF012050) was done every 2 to 3 days. To create a stable cell line with a constitutive miRFP670 nuclear marker, a PiggyBac integration plasmid pMTK62 (SV40p-H2B-miRFP670-SV40t) was constructed by modular cloning (MoClo) using a mammalian adaptation of the yeast toolkit[69]. For genomic integration, HeLa cells were transfected with Lipofectamine 2000 (Invitrogen/Thermo Fisher Scientific, Waltham, MA, US, cat. no. 11668019) at a 1:3 (μg DNA to μL Lipofectamine 2000) ratio in Opti-MEM I (Gibco/Thermo Fisher Scientific, Waltham, MA, US, cat. no. 31985062). The DNA mixture had a 5:1 pMTK62 to piggyBac transposase plasmid[70] ratio. The mixture was incubated for 10 min at room temperature before it was added to the culture. The medium was exchanged after 24 h. 72 h after transfection, the medium was exchanged again and supplemented with the 50 μg/mL selection antibiotic Hygromycin B (Invitrogen/Thermo Fisher Scientific, cat. no. 10687010). The polyclonal population was selected for 1 week with regular splitting every two days. The polyclonal population was sorted according to their fluorescence intensity with a BD FACS Aria III (Becton Dickinson, Franklin Lakes, NJ, US). The monoclonal population was expanded and then maintained in a medium with Hygromycin B.

For the co-culture optogenetic experiments, mixed cultures of CHO-K1 and HeLa cells (seeded at equal densities) were cultivated in Ham's F12 Medium as described above.

For single-cell segmentation, tracking, and illumination test experiments, the aforementioned HeLa cells were seeded in the 24-well plate (mentioned above) with DMEM medium (described above) the day prior to time-lapse imaging under the microscope.

### Sample illumination (Fig. 2 results)

Cell cultures were all-over illuminated using tailor-made panels housing 455 nm LEDs (LD W5SM, OSRAM, Munich, Germany) at an intensity of 10 μmol m$^{-2}$s$^{-1}$ for 2D, and 20 μmol m$^{-2}$s$^{-1}$ for 3D cultures for 24 h, if not indicated otherwise. Intensities were calibrated using an AvaSpec-ULS2048 fiber optic spectrometer (Avantes, Apeldoorn, Netherlands).

### Imaging and illumination system

All images were taken under a Nikon Ti2-E inverted microscope (Nikon Instruments), equipped with 4X (MRD00045), 10X (MRD00105), 20X (MRH08230), and 40X (MRH08430) objectives, all acquired from Nikon AG, Egg, Switzerland. One CMOS camera ORCA-Flash4.0 LT PLUS (Hamamatsu Photonic, Solothurn, Switzerland) was also integrated with the microscope. Following imaging set-ups were used in the microscope. Brightfield imaging, default Nikon DiaLamp with diffuser and green interference filter placed in the light path; fluorescence imaging, Spectra X Light Engine fluorescence excitation light source (Lumencor, Beaverton, USA); SYTOX Blue stain imaging, 440/20 nm LED line, HC-BS458 beam splitter, 483/32 nm emission filter; GFP imaging, 470/24 nm LED line, HC-488 LPXR beam splitter, 520/35 nm emission filter; miRFP670 imaging, 640/30 nm LED line, HC-BS660 beam splitter, 692/40 nm emission filter. All filters and beam splitters were acquired from AHF Analysetechnik AG, Tübingen, Germany.

For imaging of cell clones of CHO-K1^ApOpto^ cells and confocal imaging of 3D tissue samples, a CFI Plan Fluor 10X/NA=0.30 objective (MRH20101, Nikon, Minato City, Tokyo, Japan) was used. EGFP

epifluorescence was visualized using F49-470 470/40 ET Bandpass, F48-495 Beamsplitter T 495 LPXR, and F47-525 525/50 ET Bandpass filter elements. For confocal images, SYTOX Blue and EGFP were excited with 405 and 488 nm lasers, respectively, combined with the MHE46660 filter block (Nikon) as well as the F47-525 525/50 nm ET Bandpass, F48-559 Laser Beamsplitter H560, and the F37-593 593/40 nm Brightline HC filter elements. For the visualization of cell death, cell cultures were supplemented with 1 $\mu$M SYTOX Blue Nucleic Acid Stain (Thermo Fisher, Waltham, MA, US, cat. no. S11348) prior to imaging.

The microscope was placed inside an opaque environmental box (Life Imaging Services, Switzerland), which maintained the temperature and humidity inside at 37 °C and 95% respectively together with 5% CO2 supply. This box also shielded the cell sample from external light.

Spatial illumination was achieved via $\mu$PS framework attached with a 450 nm LED light source (M450LP2, Thorlabs). Illumination irradiance on the sample plane of the microscope was measured by using a microscope slide photodiode power sensor (S170C, Thorlabs) with a power meter interface (PM100USB, Thorlabs).

### Microscope configurations
For spatial apoptosis-patterning with CHO-K1$^{ApOpto}$ and HeLa$^{miRFPnuc}$ cells (Figs. 3 and 4), the microscope was configured with 4X objective with an additional 1.5X manual (detection) magnification. Cells were continuously illuminated via $\mu$PS framework with imaging performed every 30 min throughout the experiment. For camera capture, the following exposure times were used: 50 ms for brightfield imaging, 100 ms for reflected projection imaging, 100 ms for SYTOX Blue stain imaging, 150 ms for GFP imaging, and 500 ms for miRFP670 imaging.

During tic-tac-toe game experiments (Fig. 5), the microscope was configured with 4X objective with an additional 1.5X manual (detection) magnification. The following exposure times were used during image capture: 50 ms for brightfield imaging, 100 ms for reflected projection imaging, 300 ms for SYTOX Blue stain imaging, and 300 ms for GFP imaging. For the game in which player A won (Fig. 5c), imaging was performed every 1 h, and for the tie game (Fig. 5d), it was performed every 5 h throughout the game experiment.

For segmentation and single-cell illumination results (Fig. 1d), a 10X objective was used with no additional manual (detection) magnification. Both brightfield and reflected projection images were captured with 100 ms exposure. For tracking test experiments (Fig. 1e), 20X objective was used without any additional (detection) magnification. Brightfield and miRFP670 imagings were performed every 10 min with 150 ms and 500 ms exposure times, respectively.

### $\mu$PatternScope framework – assembly and installation
Parts list for the construction of $\mu$PS hardware is mentioned in the Supplementary Section S1. All CAD designs, relevant assembly instructions, software suite codes, and details are provided in the GitHub repository https://github.com/santkumar/uPatternScope (Zenodo DOI 10.5281/zenodo.13926233).

### Statistics and reproducibility
The cell segmentation and cell tracking experiments (shown in Fig. 1d, e) were each performed once for verification within $\mu$PS implementation. As mentioned in the Results section, the associated segmentation and tracking algorithms are adapted from previous publications[29,40], where they have been benchmarked and characterized in detail.

The qualitative verification and patterning experiments, shown in Figs. 2d, 3 and 4, were each performed once independently. Results from patterning experiments (Figs. 3 and 4) themselves reproduce the verification results shown in Fig. 2d. Patterning experiment with mixed cell cultures (CHO-K1$^{ApOpto}$ and HeLa$^{miRFPnuc}$), in Fig. 4, further validates the results from apoptosis patterning experiments in Fig. 3.

The light-controlled "apoptotic" tic-tac-toe game experiments were repeated two times independently, with different player settings. The results from these two experiments are shown in Fig. 5c, d.

### Data collection and analysis software
The following software versions were employed in this study:
- YouScope R2020 (build 2.2.0, open-source)[38] – used in the $\mu$PS software framework (Fig. 1c).
- MATLAB R2020b (academic use) – used for the $\mu$PS software development.
- Autodesk Fusion (academic license) – used for computer-aided design of the $\mu$PS hardware assembly.

Furthermore, all analyses in this study were performed on MATLAB R2020b (academic use) or Jupyter Notebook platform using custom scripts. Microscopy images were fused and formatted using MATLAB R2020b (academic use) and ImageJ 1.53t. Manuscript figures were structured and formatted on Inkscape (v0.92 and v1.2, open-source).

### Reporting summary
Further information on research design is available in the Nature Portfolio Reporting Summary linked to this article.

## Data availability
All relevant data and captured images are available in the figures. Source data are provided with this paper. DNA samples are available on request. Source data are provided with this paper.

## Code availability
All relevant hardware designs, code, and software are available on GitHub repository: https://github.com/santkumar/uPatternScope (Zenodo https://doi.org/10.5281/zenodo.13926233).

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

## Acknowledgements

We thank Dr. Stephanie Aoki for proofreading the manuscript, Dr. Oliver Hilsenbeck for providing the fastER cell segmentation source code for command line operation, Dr. Dirk Benzinger for providing the HeLa^miRFPnuc cell line, Rui Li for the projection calibration routine, and Paul Argast for hardware workshop services and for designing the microscope adapter for attaching µPS hardware with the microscope. We thank S. Kuschel and R. Schönle (University Düsseldorf) for technical support. We also thank Dr. Tom Lummen (Single Cell Facility, D-BSSE, ETH Zürich) for guiding the optical design and for help in performing uniformity test & analysis. This project was supported by a FET-Open research and innovation actions grant under the European Union's Horizon 2020 research and innovation programme (CyGenTiG; grant agreement no. 801041) to S.K., H.M.B., M.D.Z. and M.K., the German Research Foundation (DFG) under German's Excellence Strategy CEPLAS - EXC-2048/1 Project ID 390686111 to M.D.Z. and H.M.B. was supported by the 'Freigeist' Fellowship of the Volkswagen Foundation.

## Author contributions

M.D.Z. and M.K. conceived the project, secured funding, and supervised the project. S.K. and M.K. conceived the µPS framework design idea. S.K. developed the µPS hardware-software platform. H.M.B. and M.D.Z. conceived the optogenetically controlled apoptosis platform. H.M.B. engineered the optogenetic constructs and apoptosis cell lines. S.K., H.M.B. and M.C. performed the spatial patterning experiments. M.K. conceived the tic-tac-toe feedback demonstration idea. S.K. implemented and performed the feedback demonstration experiments. S.K. and H.M.B. wrote the manuscript. All authors contributed to the writing of the final draft of the manuscript.

## Funding

## Competing interests

The authors declare no competing interests.
