## [Transparent Peer Review file · Nature Communications]

Image-Guided Optogenetic Spatiotemporal Tissue Patterning using μ PatternScope

Corresponding Author: Professor Mustafa Khammash

Version 0:

Reviewer comments:

Reviewer #1

(Remarks to the Author)

The work by Kumar et al. presents an optical device, together with the software to control it, to perform optogenetic activation in biological systems. The optical device consists in a digital micromirror device (DMD) coupled to a light source and integrated in an inverted microscope. The software, written in Matlab, allows the control of light patterns and uses YouScope (plus a standalone software to send patterns to the DMD). The software suite is complemented with a module for live cell segmentation and cell tracking. The system is then applied to induce apoptosis in 2D cell cultures. The authors nicely show that they are able to control apoptosis with an excellent spatial resolution, and they performed very clean controls to assess phototoxicity. In a last part, the authors present a feedback routine where the results of the experiment are fed in the optogenetic activation routine. While the whole technical description is clearly of interest for the community, this last part is relatively disappointing given that the feedback is completely artificial and does not serve to understand any biological process.

Regarding the first technical part, I greatly appreciated the effort to open the hardware and software tools to the community. All the details are provided and documented, from the list of components (where to buy, price, etc.), the mounting procedure, to the readme of the codes. I feel being a targeted reader, and I will very likely build a similar system in the lab. In the interest of touching an even broader community, it would have been nice to see it being developed in python (open source) and together with software widely spread in the community (such as napari). Yet, having the source codes, it should be relatively simple to adapt the routines to other platforms.

On the optical part, the patterns look very crisp (eg Fig S2b) on the focal plane, on 2D samples. However, given that the DMD plane is conjugated with the specimen, I was wondering how the pattern of light diverges along the z axis. This could be of importance for thick sample, for example spheroids. This could be assessed with imaging the pattern of light in a thick (~100-200 μ m) fluorescent gel.

When working with DMD, an important quantitative data is the dynamic range, and in particular the values of the black pixels. Can the authors quantify the fold increase between the black and white squares of Fig S3 for the different objectives (after removing background of course)? This is of importance since the black regions are never completely black and could be enough to activate the optogenetic systems.

Regarding the optogenetic application, the efficiency of the system is very convincing (Fig 3a-c). Yet, given the quantitative power of the tools I found that the characterization of the system lacked quantitative analysis. Especially given the Fig 3d, where one can see that graded responses can be achieved. It would be great to know the dynamic range of the biological response as a function of the light intensity and duration of exposure. At least, the authors could quantify the data in this figure.

I was very surprised to see that the optogenetic system worked only with continuous light and not pulsatile activations. Given that the reversion time of TULIPs is 17s, I would expect that a short pulse of light every 10s or so would be enough to maintain the system bound. Even if the authors present very convincing control on the phototoxicity of the continuous illumination, the use of short pulses of light should limit the total dose received by the cells and is more compatible with standard timelapse microscopy routines. I agree that a pulse every 10s might not be enough to keep a steady state high enough, but then short (10ms) pulses every second at least should do the job, if pulses are strong enough. At least, can the

authors show the results with pulses of light in the supplementary data?

Regarding the last part on the optogenetic feedback, the authors should temperate their findings, they emphasize a lot about their approach but at the end the feedback is completely artificial and is not really a 'tour de force'.

(Remarks on code availability)

the code is nicely documented and I was able to run it. All good on my side.

Reviewer #2

(Remarks to the Author)

The manuscript describes a combination of a DMD-based illumination device and a light-sensitive cell line. The device can be controlled by an open-source software, which also controls a microscope and performs image acquisition, segmentation and tracking. The authors demonstrate in a proof of principle experiment how the system can project different patterns, causing apoptosis of the engineered cell line on selected areas.

In my opinion, the manuscript does not describe a significant advance to the field and should be considered for publication in a more specialized journal.

According to the authors, the main advantages of the system they developed are the competitive price and the flexibility of customization of the controlling software. I agree on the first point, but several commercially available options based on DMDs offer similar levels of customization. Also, the ability of the system to react to inputs (segmented images, acquired by a camera controlled by the software) has not been in my opinion fully exploited in this manuscript, since the authors use a cell line which requires several hours of exposure to undergo apoptosis, rendering the quick modulation a DMD is capable of somehow pointless. It would have been far more a compelling story if the system would have been used together with e.g. Ca⁺ imaging or other systems with a fast response.

(Remarks on code availability)

I don't have a Matlab license to test the software

Reviewer #3

(Remarks to the Author)

The study is well-planned and well-executed, but its size raises concerns about whether it should be published in two papers. The worry is that it might make it hard for readers to follow the main points of the study, as they might become diluted. To address this concern, the reviewer recommends combining the two manuscripts into one. This would make the main storyline and points more visible and easier to follow. The supplementary section can then be used to include the details that are not published in the main manuscript.

(Remarks on code availability)

Version 1:

Reviewer comments:

Reviewer #1

(Remarks to the Author)

I thank the authors for their revised manuscript and added data. I'm satisfied, two last comments:

- regarding my point on the divergence along the z axis, I was not asking for the full characterization along that axis (which requires a separate objective at 90°), but the imaging of a thick sample: is there is a large divergence, a pattern will appear blurred. Thus the amount of blur for thick sample provides an estimate of the divergence. I still think it would be good to add it, but I let the authors decide.
- for the dynamic range, as seen in the new data the fold increase (white to black) is around 3 for the checkboard while it is 1000 for full chips. Thus, there is several orders of magnitude difference. I agree that the fold increase with the checkboard depends on many parameters, but when a pattern is shined on cells the situation will be closer to the checkboard than the full chip. Thus, one can expect a significant amount of light on the dark regions of the projected pattern, which could call for a delicate tuning of the white/black intensities to achieve the opto activation only within the pattern. I would add a sentence to warn the reader on this risk.

(Remarks on code availability)

all good

Reviewer #2

(Remarks to the Author)

In the rebuttal letter, the authors clarify the advantages of their integrated approach and justify the choice of biological system

they used in their study. They also acknowledge the availability of commercial solutions which, albeit dependant on the specific hardware, could be used to perform similar functions. While having a more integrated system could be an advantage, I still wonder whether this alone represents a novelty and justifies publishing in a journal targeted at a general audience.

(Remarks on code availability)

Response to Reviewers

Optogenetic Feedback with μ PatternScope: A New Paradigm for Dynamic Spatiotemporal Tissue Patterning

Sant Kumar, Hannes M. Beyer, Mingzhe Chen, Matias D. Zurbriggen, Mustafa Khammash

We appreciate the editor's evaluation of our work for further consideration, and we are thankful to the three reviewers for their assessments and valuable feedback. Their suggestions have strengthened our manuscript and enhanced its clarity. Based on the reviewers' comments, relevant modifications have been incorporated in the revised manuscript. These changes have been highlighted in blue into the manuscript text.

Below are the reviewers' comments and our point-by-point response (highlighted in blue).

Reviewer - 1

The work by Kumar et al. presents an optical device, together with the software to control it, to perform optogenetic activation in biological systems. The optical device consists in a digital micromirror device (DMD) coupled to a light source and integrated in an inverted microscope. The software, written in Matlab, allows the control of light patterns and uses YouScope (plus a standalone software to send patterns to the DMD). The software suite is complemented with a module for live cell segmentation and cell tracking. The system is then applied to induce apoptosis in 2D cell cultures. The authors nicely show that they are able to control apoptosis with an excellent spatial resolution, and they performed very clean controls to assess phototoxicity. In a last part, the authors present a feedback routine where the results of the experiment are fed in the optogenetic activation routine. While the whole technical description is clearly of interest for the community, this last part is relatively disappointing given that the feedback is completely artificial and does not serve to understand any biological process.

We thank the reviewer for her/his positive feedback on our work and thorough assessment of the manuscript. In this manuscript, we indeed focus on the technical aspects of the μ PatternScope framework, and highlight technological aspects and related details to the optogenetics research community. The last part of our paper intends to demonstrate the technical feasibility of achieving *in silico* control of an underlying biological design using the proposed system.

Regarding the first technical part, I greatly appreciated the effort to open the hardware and software tools to the community. All the details are provided and documented, from the list of components (where to buy, price, etc.), the mounting procedure, to the readme of the codes. I feel being a targeted reader, and I will very likely build a similar system in the lab. In the interest of touching an even broader community, it would have been nice to see it being developed in python (open source) and together with software widely spread in the community (such as napari). Yet, having the source codes, it should be relatively simple to adapt the routines to other platforms.

We thank the reviewer for this notable suggestion. YouScope (open source microscope control software) is one of the central aspects of our software routine, and currently, it has been implemented for MATLAB script integration. Since MATLAB's academic license is readily available to almost every academic institution, we decided to develop our software framework based on it. With the emergence of cross-platform coding tools and codecs for converting a given code to different programming languages, we believe that it will be possible for individual labs to adapt the codebase for integration into their preferentially-used framework. In addition, python code for controlling the DMD device (not the microscope) exists, for example, in the Pycrafter6500 GitHub repository (<https://github.com/csi-dcsc/Pycrafter6500>) which might support the development of python-based framework alternatives.

On the optical part, the patterns look very crisp (eg Fig S2b) on the focal plane, on 2D samples. However, given that the DMD plane is conjugated with the specimen, I was wondering how the pattern of light diverges along the z axis. This could be of importance for thick sample, for example spheroids. This could be assessed with imaging the pattern of light in a thick (100-200 μm) fluorescent gel.

As shown in the above representative light path diagram of the μ PatternScope hardware attached to the microscope, the DMD pattern projection (in blue) and the camera detection (in red) light paths share the same microscope objective lens. Moving the microscope objective along the z-axis will not only change the detection focus for the camera (or eyepiece) but it will also change the focus of DMD pattern projection. One way to observe the DMD projected pattern characteristics along the z-axis is to employ a separate detection objective at a 90-degree angle to the microscope objective. This is not possible to implement on the Nikon Ti2-E microscope that we used in this work.

When working with DMD, an important quantitative data is the dynamic range, and in particular the values of the black pixels. Can the authors quantify the fold increase between the black and white squares of Fig S3 for the different objectives (after removing background of course)? This is of importance since the black regions are never completely black and could be enough to activate the optogenetic systems.

This is an important point. We thank the reviewer for this suggestion. We have now included fold increase values between black and white squares in Supplementary Figure S3 (a). However, we would like to mention that the fold increase in pixel intensity values from black to white squares (in the checkerboard pattern projection) depends on several factors. Apart from the DMD light illumination intensity, camera parameters (e.g. exposure time, binning strategy, etc.), any additional magnification in the camera capture (detection) pathway of the microscope, projection pattern (e.g. size of black and white squares in a checkerboard pattern), and microscope slide (where the pattern is being projected) properties (e.g. thickness, reflectivity, etc.) also affect pixel intensities captured by the microscope camera. For a more representative fold increase value between black and white illumination pattern, we have captured full field of view black and white image projections. We have now shown the fold increase values calculated from mean pixel intensities from those images in Supplementary Figure S3 (b).

Regarding the optogenetic application, the efficiency of the system is very convincing (Fig 3a-c). Yet, given the quantitative power of the tools I found that the characterization of the system lacked quantitative analysis. Especially given the Fig 3d, where one can see that graded responses can be achieved. It would be great to know the dynamic range of the biological response as a function of the light intensity and duration of exposure. At least, the authors could quantify the data in this figure.

We thank the reviewer for pointing this out. We have now quantified the data in Figure 3d and show the results in Supplementary Figure S8.

I was very surprised to see that the optogenetic system worked only with continuous light and not pulsatile activations. Given that the reversion time of TULIPs is 17s, I would expect that a short pulse

of light every 10s or so would be enough to maintain the system bound. Even if the authors present very convincing control on the phototoxicity of the continuous illumination, the use of short pulses of light should limit the total dose received by the cells and is more compatible with standard timelapse microscopy routines. I agree that a pulse every 10s might not enough to keep a steady state high enough, but then short (10ms) pulses every second at least should do the job, if pulses are strong enough. At least, can the authors show the results with pulses of light in the supplementary data?

We have now included the new experimental results with pulsatile illumination strategy in Supplementary Figure S9. Indeed, a pulsatile illumination regime with a 50% duty cycle (0.5 s in 1 s) successfully induced the gene switch and was able to reduce phototoxic effects.

Regarding the last part on the optogenetic feedback, the authors should temperate their findings, they emphasize at lot about their approach but at the end the feedback is completely artificial and is not really a ‘tour de force’.

We thank the reviewer for this critical assessment. We have now revised several statements in the optogenetic feedback part as well as the discussion part of the manuscript. As mentioned previously, our manuscript sets a primary focus on the technical aspects of the proposed μ PatternScope framework, aiming to describe its technological capabilities and details relevant to the optogenetics research community. The last part of our paper intends to demonstrate the technical feasibility of achieving *in silico* control of an underlying biological design using the developed system.

Reviewer - 2

The manuscript describes a combination of a DMD-based illumination device and a light-sensitive cell line. The device can be controlled by an open-source software, which also controls a microscope and performs image acquisition, segmentation and tracking. The authors demonstrate in a proof of principle experiment how the system can project different patterns, causing apoptosis of the engineered cell line on selected areas.

In my opinion, the manuscript does not describe a significant advance to the field and should be considered for publication in a more specialized journal.

According to the authors, the main advantages of the system they developed are the competitive price and the flexibility of customization of the controlling software. I agree on the first point, but several commercially available options based on DMDs offer similar levels of customization. Also, the ability of the system to react to inputs (segmented images, acquired by a camera controlled by the software) has not been in my opinion fully exploited in this manuscript, since the authors use a cell line which requires several hours of exposure to undergo apoptosis, rendering the quick modulation a DMD is capable of somehow pointless. It would have been far more a compelling story if the system would have been used together with e.g. Ca⁺ imaging or other systems with a fast response.

We appreciate the critical evaluation of our manuscript by the reviewer. Although it is correct that commercially available hardware options based on DMD technologies exist (e.g. Mosaic3 from Andor), they still require customizations to be developed by the user which are very specific to the existing software framework provided by the manufacturer (e.g. in the form of plugins). While the software solutions of DMD devices might allow automatization and programmatic interfaces, operating the device seamlessly together with an automated microscope governed by a controller software will be challenging if possible at all. As far as we know, no standard software toolkit incorporates DMD control, microscope control, and computational plugins (e.g. cell segmentation, tracking, etc.) with those commercial options. With the μ PatternScope framework, we provide, at a fraction of the cost and open source, a complete package encompassing DMD-based hardware and software controls, which seamlessly integrates DMD projection control with microscope control and which may freely be expanded. Besides that, we also integrate an example suite encompassing computational tools such as cell segmentation, which may be utilized or adapted for specific applications.

Regarding our choice of application, we chose gene expression control as it reflects our main application interest. While associated with a relatively slow response time intrinsic to the mechanisms governing protein synthesis and degradation, optogenetic gene expression regulation remains highly flexible for various potential applications. However, as stated by the reviewer, μ PatternScope can also operate at

much faster time scales, as demonstrated in this manuscript (Figure 1d). In fact, the pattern switching time takes about 2-3 seconds (sequence of patterns can be pre-defined for close to real-time switching). The complete programmatic routine run to get Figure 1d results involving image acquisition, image processing (cell segmentation), and image pattern projection can be completed in about 5-6 seconds, assuming the camera exposure time of 500 ms with a single channel.

Reviewer - 3

The study is well-planned and well-executed, but its size raises concerns about whether it should be published in two papers. The worry is that it might make it hard for readers to follow the main points of the study, as they might become diluted. To address this concern, the reviewer recommends combining the two manuscripts into one. This would make the main storyline and points more visible and easier to follow. The supplementary section can then be used to include the details that are not published in the main manuscript.

We thank the reviewer for examining both manuscripts and for providing a favorable assessment. We decided to draft two separate manuscripts, as we see two distinct areas of interest for the community in our work. While one manuscript deals with the genomic engineering of cells to equip them with optogenetic gene switches and explores their potential to control 2D and 3D cellular processes, the other manuscript centers on the development and integration of the μ PatternScope hardware and software into optogenetic biological experiments.

The valuable feedback received from the reviewers during the revision process strengthens our notion of this strategy. Reviewer - 1 of this manuscript identified her/himself as a ‘targeted reader’ likely to implement a similar system, and it could be well possible that she/he aims to extend the utility of the device beyond mammalian cell systems. This feedback – as we think – shows that there is a striking relevance of our developed illumination technology that generates interest within the target audience. Conversely, the primary readership for the other manuscript may not have a particular interest in the technical details of μ PatternScope (in fact, Reviewer - 1 encouraged us to consider the removal of technical DMD description unrelated to the biological implementation from the other manuscript, which we did). Our focus in the other manuscript remains on developing gene switches compatible with diverse illumination strategies, serving a broad spectrum of applications for spatiotemporal control of cells and tissues in 2D and 3D. We hope that Reviewer - 3 will agree with our rational argumentation and appreciate the value of addressing distinct groups of target readers.

Response to Reviewers

Image-Guided Optogenetic Spatiotemporal Tissue Patterning using μ PatternScope

Sant Kumar, Hannes M. Beyer, Mingzhe Chen, Matias D. Zurbriggen, Mustafa Khammash

Many thanks to the editor and reviewers for their consideration and helpful feedback in further improving our manuscript.

Here are the reviewers' comments and our point-by-point response (highlighted in blue).

Reviewer - 1

I thank the authors for their revised manuscript and added data. I'm satisfied, two last comments:

We appreciate the reviewer's positive assessment of our updated manuscript.

Regarding my point on the divergence along the z axis, I was not asking for the full characterization along that axis (which requires a separate objective at 90°), but the imaging of a thick sample: is there is a large divergence, a pattern will appear blurred. Thus the amount of blur for thick sample provides an estimate of the divergence. I still think it would be good to add it, but I let the authors decide.

We thank the reviewer for this impactful suggestion. We believe that this indeed would be a useful information to add, albeit in a separate manuscript addressing optogenetic stimulation in 3D-samples. The focus of μ PatternScope framework design and its application, as presented in this manuscript, has been inducing patterns over 2D-cell cultures. Characterization along z-axis would require creating a stable multi-layered cell culture, and engineering a suitable optogenetic system with higher induction light wavelengths allowing considerable penetration into 3D-samples. This additional work would significantly shift the emphasis of the current manuscript.

For the dynamic range, as seen in the new data the fold increase (white to black) is around 3 for the checkboard while it is 1000 for full chips. Thus, there is several orders of magnitude difference. I agree that the fold increase with the checkboard depends on many parameters, but when a pattern is shined on cells the situation will be closer to the checkboard than the full chip. Thus, one can expect a significant amount of light on the dark regions of the projected pattern, which could call for a delicate tuning of the white/black intensities to achieve the opto activation only within the pattern. I would add a sentence to warn the reader on this risk.

We thank the reviewer for further expanding on this suggestion. We have now added these sentences in the Discussion section of the manuscript: Depending on the targeting pattern and application requirements (for example, targeting single-cells in a confluent cell culture), one might need to tune light intensities within the projection pattern to ensure minimal illumination bleed-through between light and dark regions or between neighboring cells (Supplementary Figure S3). Solutions, such as erosion of light intensities within the stimulation region [Fox et. al., Nature Communications 2022], can be easily implemented in our software framework.

Reviewer - 2

In the rebuttal letter, the authors clarify the advantages of their integrated approach and justify the choice of biological system they used in their study. They also acknowledge the availability of commercial solutions which, albeit dependant on the specific hardware, could be used to perform similar functions. While having a more integrated system could be an advantage, I still wonder whether this alone represents a novelty and justifies publishing in a journal targeted at a general audience.

Thank you for your feedback. We appreciate your acknowledgment of our efforts to clarify the benefits of our integrated system and to contextualize it within existing commercial solutions. While it is true that some commercial alternatives can achieve similar isolated functions, μ PatternScope represents a significant advancement due to its unique combination of accessibility, adaptability, and integrated control, making it highly relevant for a broader scientific audience. Specifically:

1. **Accessibility and Flexibility:** Unlike commercial solutions tied to specific proprietary hardware, μ PatternScope is designed to be adaptable across various experimental setups, which increases accessibility for researchers in diverse settings. This flexibility makes it an attractive option for laboratories where budget constraints or equipment compatibility limit the use of commercial alternatives. By providing an open-source, modular framework with an affordable hardware component, μ PatternScope is accessible to a wide range of researchers, including those in resource-limited environments.
2. **Novelty of Integration and “Smart Microscopy”:** μ PatternScope stands out by consolidating functionalities typically distributed across different platforms, enabling dynamic and automated control based on real-time cellular imaging data. This capability supports “smart microscopy,” allowing for adaptive adjustment of experimental conditions directly in response to cellular changes observed in real-time. This degree of control, integrated within a single framework, simplifies complex workflows, enhances experimental reproducibility, and is difficult to achieve with existing systems.
3. **Generalization and Potential Impact:** Beyond the specific biological system demonstrated in this study, μ PatternScope’s capabilities are highly generalizable. The system’s spatiotemporal control and adaptable software architecture support a broad range of applications, from dynamic tissue patterning to other areas in tissue engineering, synthetic biology, and systems biology. We believe this versatility makes μ PatternScope a valuable tool for advancing research in fields that require precise control over cellular processes, aligning well with the journal’s mission to address a general scientific audience.

In summary, μ PatternScope’s unique advantages—including its accessible, integrated approach and potential for real-time adaptability—address current limitations in optogenetic control and microscopy. We hope this response clarifies how our system not only enhances existing capabilities but also represents a meaningful innovation for diverse applications.